# Positive Correlation of the Gene Rearrangements and Evolutionary Rates in the Mitochondrial Genomes of Thrips (Insecta: Thysanoptera)

**DOI:** 10.3390/insects13070585

**Published:** 2022-06-27

**Authors:** Qiaoqiao Liu, Jia He, Fan Song, Li Tian, Wanzhi Cai, Hu Li

**Affiliations:** 1MOA Key Lab of Pest Monitoring and Green Management, Department of Entomology, College of Plant Protection, China Agricultural University, Beijing 100193, China; 2323@cau.edu.cn (Q.L.); hejiayc@126.com (J.H.); fansong@cau.edu.cn (F.S.); ltian@cau.edu.cn (L.T.); caiwz@cau.edu.cn (W.C.); 2Institute of Plant Protection, Academy of Ningxia Agriculture and Forestry Science, Yinchuan 750002, China

**Keywords:** mitochondrial genome, gene rearrangement, Thysanoptera

## Abstract

**Simple Summary:**

*Aeolothrips*, commonly known as banded thrips, is the largest genus of the family Aeolothripidae (predatory thrips). In the current study, we sequenced the mitochondrial genome (mitogenome) of the banded thrip species *Aeolothrips xinjiangensis*. We found a novel gene arrangement in this mitogenome that has not been reported in Thysanoptera. By comparing the gene order and rearrangement patterns, we found seven identical gene blocks and three identical rearrangement events in two mitogenomes of banded thrips. There was marked variation in the mitochondrial gene order across thrip species, with only two conserved gene blocks shared by all 14 thrips. In addition, we found a positive correlation between the degree of gene rearrangement and evolutionary rate. Our results suggested that the mitogenomes of thrips have tended to be stable since their massive rearrangement.

**Abstract:**

Extensive gene rearrangement is characteristic in the mitogenomes of thrips (Thysanoptera), but the historical process giving rise to the contemporary gene rearrangement pattern remains unclear. To better understand the evolutionary processes of gene rearrangement in the mitogenomes of thrips, we sequenced the mitogenome of the banded thrip species *Aeolothrips xinjiangensis*. First, we found a novel mitochondrial gene order in this species. This mitogenome is 16,947 bp in length and encodes the typical 37 coding genes (13 protein-coding genes, 22 tRNA genes, and two rRNA genes) of insects. The gene arrangement was dramatically different from the putative ancestral mitogenome, with 26 genes being translocated, eight of which were inverted. Moreover, we found a novel, conserved gene block, *trnC-trnY*, which has not been previously reported in the mitogenomes of thrips. With this newly assembled mitogenome, we compared mitogenome sequences across Thysanoptera to assess the evolutionary processes giving rise to the current gene rearrangement pattern in thrips. Seven identical gene blocks were shared by two sequenced banded thrip mitogenomes, while the reversal of *ND2* combined with TDRL events resulted in the different gene orders of these two species. In phylogenetic analysis, the monophyly of the suborders and families of Thysanoptera was well supported. Across the gene orders of 14 thrips, only two conserved gene blocks, *ATP8-ATP6* and *ND4-ND4L*, could be found. Correlation analysis showed that the degree of gene rearrangement was positively correlated with the non-synonymous substitution rate in thrips. Our study suggests that the mitogenomes of thrips remain stable over long evolutionary timescales after massive rearrangement during early diversification.

## 1. Introduction

The metazoan mitogenome is a circular DNA that contains 13 protein-coding genes (PCGs), 22 transfer RNA genes (tRNAs), and 2 ribosomal RNA genes (rRNAs) [1]. Most insects share the same gene order, which is known as the ancestral insect mitogenome [2]. However, rapid growth of insect mitogenomic data from all insect orders has revealed that rearrangement of mitochondrial genes is common across insects, and patterns of rearrangement can be highly variable. In Hemiptera, gene rearrangement is restricted to tRNA genes (e.g., *trnI* or *trnT*) [3], whereas in the order Psocodea, rearrangement involves the translocation of protein-coding genes [4]. In Phthiraptera, intense gene rearrangement (only one to three conserved gene blocks shared with putative ancestral insects) can take place, with some species having highly fragmented mitogenomes (e.g., *Pediculus humanus* have 18 minichromosomes) [5,6,7]. The mitogenome of most thrips generally have three to eight conserved gene blocks (each block contains two to seven genes), which display the putative ancestral gene order. Outside these conserved blocks, the gene order varies dramatically across thrip species [8,9,10].

To date, four models have been proposed to explain the mechanism underlying mitochondrial gene rearrangement: (i) tandem duplication followed by the random loss (TDRL) model [11,12]; (ii) illicit priming of replication by tRNA genes [13]; (iii) recombination [14]; and (iv) duplication followed by non-random loss [15]. The TDRL model is well-accepted in studies of the gene rearrangement mechanism of many insects [16]. In thrips, gene rearrangement is a complex phenomenon often associated with gene duplication and gene loss, and often taking place in multiple control regions, making it difficult to explain the underpinning mechanism using a single model [17,18]. The ancestral state reconstructions performed by previous studies have indicated that gene rearrangements are common in thrips, and 71 derived rearrangements are shared between at least two species, while 24 of these are likely synapomorphies for phylogenetic clades [9]. However, gene rearrangements in thrips appear highly frequent and random, making phylogeny directly inferred from rearrangement data highly discordant with those reconstructed from other markers. Given the rapid rate of evolution of the mitogenome structure, thrips are an ideal group for studying the mechanisms of gene rearrangement. 

The order Thysanoptera (with more than 6000 extant species) contains two suborders, namely, Tubulifera and Terebrantia. The suborder Tubulifera has only one family, Phlaeothripidae, while Terebrantia has seven families [19]. The family Aeolothripidae is commonly known as the banded thrip. Unlike most thrips, which are plant feeders, banded thrips are facultative predators feeding on small arthropods [20,21,22], making them a potential biological control agent against agricultural pests. Previous studies of the mitogenomes in Thysanoptera mainly focused on phytophagous thrips. To date, whole mitogenomes have been sequenced in 13 thrips, only one of which, *Franklinothrips vespiformis* (Aeolothripidae), is predatory. In the current study, we sequenced the mitogenome of *Aeolothrips xinjiangensis* to illustrate more mitogenomic features of predatory thrips. The gene arrangement traits of banded thrips were also summarized and compared to other thrips. Based on a reconstructed phylogenetic tree, we also investigated the correlation between the degree of gene rearrangement and evolutionary rate. Our study sheds light on the mitogenomic structure of predatory thrips and paves the way to understanding the evolution and mechanism of gene rearrangements in Thysanoptera. 

## 2. Materials and Methods

### 2.1. Sampling and Mitogenome Acquisition

Specimens of *Aeolothrips xinjiangensis* [23] were collected from Gouji in Shizuishan, Ningxia, China. Living thrips were soaked in absolute ethyl alcohol and stored at −20 °C. Total genomic DNA was extracted from two adult thrips using a DNeasy Blood and Tissue kit (Tiangen, Beijing, China) following the manufacturer’s instructions. A 610 bp *COI* gene fragment of the mitogenome was amplified using the primer pairs LCO1490 (5′-GGTCAACAAATCATAAAGATATTGG-3′) and HCO2198 (5′-TAAACTTCAGGGTGACCAAAAAATCA-3′) [24]. The PCR reaction mix included 25 μL aliquots: 1 μL template, 1 μL 10 nmol/μL primers, 9.5 μL ultra-pure water, and 12.5 μL 2× Premix (EmeraldAmp Max PCR master Mix, TaKaRa, Dalian, China). The PCR cycles were carried out as follows: 94 °C for 20 s, 94 °C for 1 min; 35 cycles of 94 °C for 1 min, 48 °C for 50 s, 72 °C for 1 min, and an extension step of 72 °C for 10 min. The PCR products were purified and sequenced by Sangon Biotechnology Co., Ltd. (Shanghai, China).

An Illumina TruSeq library was prepared with an average insert size of 350 bp and was sequenced using the Illumina Hiseq 2500 platform with 150 bp paired-end reads. Adapters were trimmed from raw reads using Trimmomatic [25]. Low-quality and short reads were removed with Prinseq [26]. High-quality reads were used in de novo assembly with IDBA-UD [27], with minimum and maximum *k* values of 45 and 145 bp, respectively. Sequences of the *COI* (610 bp) fragment were used to identify mitochondrial assemblies using BLAST searches with at least 98% similarity. To confirm the accuracy of the assembly, clean reads were then mapped onto the obtained mitogenomic contig using Geneious 10.1.3 (http://www.geneious.com/, accessed on 15 September 2019), with mismatches of up to 2%, a maximum gap size of 3 bp, and a minimum overlap of 100 bp [28]. Finally, the complete mitogenome of *Aeolothrips xinjiangensis* was obtained with a 242× average sequencing depth.

### 2.2. Mitogenome Annotation and Analysis

The obtained mitogenome was preliminarily annotated by MitoZ [29]. Gene boundaries were confirmed by using the MITOS Web Server [30] and tRNAScan-SE Search Server [31], as well as via manual alignment with homologous genes from available mitogenomes of other thrips. The secondary structures of 20 tRNAs were predicted by tRNAScan-SE Search Server [31], and the *trnS1* and *trnT* were confirmed by their anticodon arm. Mfold Web Server [32] and Tandem Repeats Finder [33] were used to predict the stem–loop and tandem repeats within putative control regions with default parameters, respectively. 

We used the breakpoints to assess the rate of gene rearrangement between thrip and *Alloeorhynchus bakeri* [34]. This calculated the minimum number of breakpoints that must be introduced into one genome to change it to the other genome, with more breakpoints indicting the higher rates of gene rearrangement [35]. The data were the gene orders generated from the annotated mitogenomes, and the breakpoints were calculated by CREx [36]. The synonymous substitutions (Ks) and non-synonymous substitutions (Ka) were calculated based on the sequences of 13 PCGs (see the phylogenetic analysis below) by using DnaSP [37]. 

### 2.3. Phylogenetic Analysis

The mitogenomes of 14 thrip species were used for phylogenetic analysis with the damsal bug *Alloeorhynchus bakeri* as the outgroup (Table 1). Each PCG was individually aligned based on codon-based multiple alignments using the MAFFT algorithm [38] within the TranslatorX online platform [39]. Two datasets were generated: (i) PCG, which contains all codons of 13 PCGs, and (ii) PCG12, which contains the first and second codons of 13 PCGs. Many previous studies have shown that the site-heterogeneous mixture CAT + GTR model can effectively reduce the systematic errors caused by sequence heterogeneity [40,41,42]. Therefore, the phylogeny of thrips was inferred from two datasets under the CAT + GTR model using PhyloBayes MPI 1.5a [43]. Two independent Markov chain Monte Carlo chains (MCMCs) were run after the removal of constant sites from the alignment and were stopped after the two runs had satisfactorily converged (maxdiff < 0.3). A consensus tree was computed from the remaining trees combined from two runs after the initial 25% of the trees of each run were discarded as burn-in. 

## 3. Results and Discussion

### 3.1. Mitogenome Feature and Composition

The mitogenome of *Aeolothrips xinjiangensis* was a single circular molecule of 16,947 bp in length (Figure 1). The nucleotide composition of the whole genome was biased to adenine and thymine, with an A + T content of 74.3%. It contained 37 encoding genes, which usually exist in the metazoan mitogenome [44], and two putative control regions (CR1 and CR2). Seven pairs of neighboring genes had 1–4 bp overlaps (Appendix A). Twenty tRNAs formed a typical cloverleaf secondary structure, while *trnS1* lost its DHU arm and *trnT* lost its TψC arm (Appendix A). The absence of the TψC arm or DHU arm is ubiquitous in the mitogenomes of thrips [9]. The majority strand (J-strand) encodes 29 genes, whereas the minority strand (N-strand) encodes 8 genes (*trnP* and the gene block *ND2–trnC–trnY**–ND5**–trnH**–ND4**–ND4L*). 

Twelve mitochondrial PCGs of *Aeolothrips xinjiangensis* were initiated with ATN, while *COIII* used TTG as a start codon. Eleven PCGs used the canonical triplet stop codon TAA or TAG. However, *ND1* used an incomplete termination codon TA, and *ATP6* ended with a single T. The incomplete termination codons of PCGs frequently occur in thrips and other insects, completed by polyadenylation after excision of the downstream tRNA gene [45,46,47]. Except CGC, the other 63 codons of the amino acid codon table were shown to be utilized by *Aeolothrips xinjiangensis* (Appendix A).

### 3.2. Non-Coding Regions

There were 2350 non-coding nucleotides in the mitogenome of *Aeolothrips xinjiangensis*: 252 bp in 15 small intergenic regions and 2098 bp in three large non-coding regions (Appendix A). The three large non-coding regions were 1594 bp, 356 bp, and 148 bp in length. The largest non-coding region is usually considered to be the CR in the mitogenome, and its length varies between species, from 70 bp in *Ruspolia dubia* [48] to 4601 bp in *Drosophila melanogaster* [49]. We searched for specific motifs included in typical CRs among the three large non-coding regions [50]. The canonical features of the control region (T(A)n motif, G(A)nT motif, (TA)n motif, T-stretch, tandem repeats, and stem–loop) were found in the non-coding regions of 1596 bp (Figure 2A) and 356 bp (Figure 2B). We proposed that these two regions are CRs. However, the arrangement of these motifs was different between the two CRs, and more compact in CR2 (356 bp). The tandem repeat unit of CR2 was 7 bp in length and repeated six times, and the sixth copy was only 6 bp with the last base lost. In CR1 (1596 bp), the repetition unit was 36 bp, which was repeated three times, with only the initial 9 bp contained in the last repeat unit. The putative control region has been considered as the major regulation region of mitochondrial replication and transcription [1]. The tandem repeat in the control region is thought to be involved in the termination of transcription by nature of their complex secondary structures [51] or to contain the repeated transcription initiation sites [46]. The different tandem repeats may be responsible for the expression levels of different genes. 

The 148 bp non-coding nucleotides are located between *ND4* and *ND4L* in the mitogenome of *Aeolothrips xinjiangensis*. In other thrip mitogenomes, 10 species have 1–21 bp overlaps (most were 7 bp) between *ND4* and *ND4L*, and three species have 5–18 bp non-coding nucleotides (Appendix A). In the mitogenomes of insects, *ND4–ND4L* was transcribed to polycistronic mRNA with overlaps or no intergenic spacer between them [52,53,54]. Such a long intergenic spacer may divide this polycistronic mRNA into two monocistron mRNA. 

### 3.3. Gene Rearrangements in the Mitogenome of Banded Thrips 

The mitochondrial gene arrangement of *Aeolothrips xinjiangensis* displayed some novel patterns, which have never been detected among other thrips (Figure 3A). Compared to the putative ancestral gene arrangement for hexapods, the scenario for gene rearrangement in *Aeolothrips xinjiangensis* was deduced by CREx [36]. This scenario contained eight events: one reverse transposition, three reversals, and four TDRLs. At least 26 genes were translocated (changes in the relative location), and 8 of these genes (4 tRNA genes, 2 PCGs, and 2 rRNA genes) were also reversed (changes in the encoding direction) (Figure 3B). Only four ancestral gene blocks were present in *Aeolothrips xinjiangensis*: *ATP8**–ATP6*, *trnC**–trnY*, *ND5**–trnH**–ND4**–ND4L**–trnT**–trnP**–ND6*, and *trnV**–lrRNA*. *trnC**–trnY* is a novel gene block in thrips. 

According to previous studies, the gene rearrangement in *Franklinothrips vespiformis* has two alternative scenarios, each of which includes seven events: one reverse transposition, two reversals, and four TDRLs [9]. Seven identical gene blocks, including 32 genes (Figure 3A), were shared between *Aeolothrips xinjiangensis* and *Franklinothrips vespiformis*. In comparison to the putative ancestral gene order, the degree of gene rearrangement was the same in the two banded thrip species (both had 28 breakpoints in 37 encoding genes), but the rearrangement events were not exactly identical. The reverse transposition (*trnQ*) and reversals (*trnF* and *ND1**–trnL1**–lrRNA**–trnV**–srRNA*) were shared in the banded thrips. The TDRL events combined with the reversal of *ND2* resulted in different mitochondrial gene arrangements of the two banded thrips. We propose that the identical rearrangement events might have occurred in any common ancestral mitogenome of the two banded thrip species, while reversal of *ND2* in *Aeolothrips xinjiangensis* might have originated from the most recent one. 

### 3.4. Correlation between the Degree of Gene Rearrangement and the Evolutionary Rate

To better understand the evolution process of the gene rearrangement in Thysanoptera, we reconstructed a phylogenetic tree of thrips based on nucleotide sequences of protein-coding mitochondrial genes (Figure 4A). The resulting topology of the thrip phylogenetic tree is concordant with previous studies, with well-supported monophyly of all suborders and families [9,55,56]. In Terebrantia, the family Aeolothripidae was the first branching lineage, congruent with previous fossil evidence [55], whereas the family Stenurothripidae was placed as a sister group of Thripidae. Within Thripidae, the subfamily Panchaetothripinae was a sister to the remaining three subfamilies. The subfamily Thripinae remained polyphyly as the genus *Scirtothrips* was clustered with the genus *Neohydatothrips* (Sericothripinae). The sister relationship of *Scirtothrips* and *Neohydatothrips* was also recovered by morphological study, and the monophyly of Thripinae remains controversial [56].

Thrip mitogenomes are well known for their extensive gene rearrangements. In addition, in most thrips, rearrangement takes place more frequently in tRNAs than in protein-coding genes [8,17]. The complexity of gene rearrangement makes it difficult to explore the evolutionary pattern of gene arrangement based on phylogeny. Since highly rearranged tRNA genes (minor genes) may mask the conservation of protein-coding genes and rRNA genes (major genes) [57,58], we removed the minor genes to retrieve the rearrangement of major genes in thrips. Across the gene orders of 14 thrips, we identified six “mitotypes” [57] (Figure 4B). All mitotypes had rearranged when compared to the putative ancestral gene order (mitotype 0). Only two conserved gene blocks, *ATP8–ATP6* and *ND4–ND4L,* could be found in all mitotypes. The gene block *ND1**–COI–COII–ATP8–ATP6* was shared by the mitotypes of Tubulifera (mitotypes 1 and 2), while *ND5–ND4–ND4L–ND6–lrRNA–COI* was only detected in the four mitotypes of Terebrantia. The translocation of *srRNA* and the inversion of *ND2* happened between two mitotypes (mitotypes 3 and 4) of Aeolothripidae. Within Thripidae, there were two mitotypes (mitotypes 5 and 6) and the species of mitotype 6 were clustered in one clade. The only difference between mitotypes 5 and 6 was the translocation of *ND3* (*COII–COIII–ND3* to *ND3–COII–COIII*). In terms of the encoding direction of the genes, t10 genes were reversed between the two mitotypes of Tubulifera, while only the inversion of *ND2* was found among the four mitotypes of Terebrantia.

The rate of non-synonymous substitutions (Ka) and breakpoints are good parameters for measuring the evolutionary rate and degree of gene rearrangement [34,35]. The correlation between the degree of gene rearrangement and evolutionary rates of the mitogenome were tested for each thrip mitogenome using *Alloeorhynchus bakeri* as a reference (Appendix A). Compared to Terebrantia, the breakpoints, rate of non-synonymous substitutions, and the branch lengths of the phylogenetic tree in Tubulifera were obviously higher (Figure 4A). In the normal test, the breakpoints and Ka values followed the normal distribution, while the branch lengths did not (Appendix A). Correlation analysis showed that the degree of gene rearrangement in thrip mitogenomes is positively correlated with the rate of non-synonymous substitution and the branch length (Table 2). A previous study also showed a positive correlation between gene rearrangement rates and nucleotide substitution in hemipteraoid [35], whereas no correlation was detected in Heteroptera (Hemiptera) [3]. Our study indicates that the degree of gene rearrangement and the nucleotide substitution rate in mitogenomes are positively correlated in the taxa with extensive gene rearrangements. We observed higher breakpoint and Ka values in Tubulifera and Aeolothripinae, contrasting the relatively low and stable values among other Thripidae species. Such results suggest that the mitogenomes of thrips may remain stable over long evolutionary timescales after some massive rearrangement during the early speciation stage. A denser taxonomic sampling for mitogenomes, especially in the suborder Tubulifera, would allow a better understanding of the pattern and history of the evolution of Thysanoptera mitogenomes. 

## 4. Conclusions

Thrips display exceptional interspecific variation in mitogenomic structure, making them an ideal model for studying mitochondrial evolution. The mitogenome of *Aeolothrips xinjiangensis*, reported in the current study, provided a new case of mitochondrial gene arrangement in thrips. The intensification of gene rearrangement may relate to an elevated mitochondrial substitution rate. Our results facilitate a more robust phylogenetic reconstruction for the order Thysanoptera and provide further clues for understanding the history and driving mechanisms of mitochondrial gene rearrangement in this insect’s lineage.

## Figures and Tables

**Figure 1 insects-13-00585-f001:**
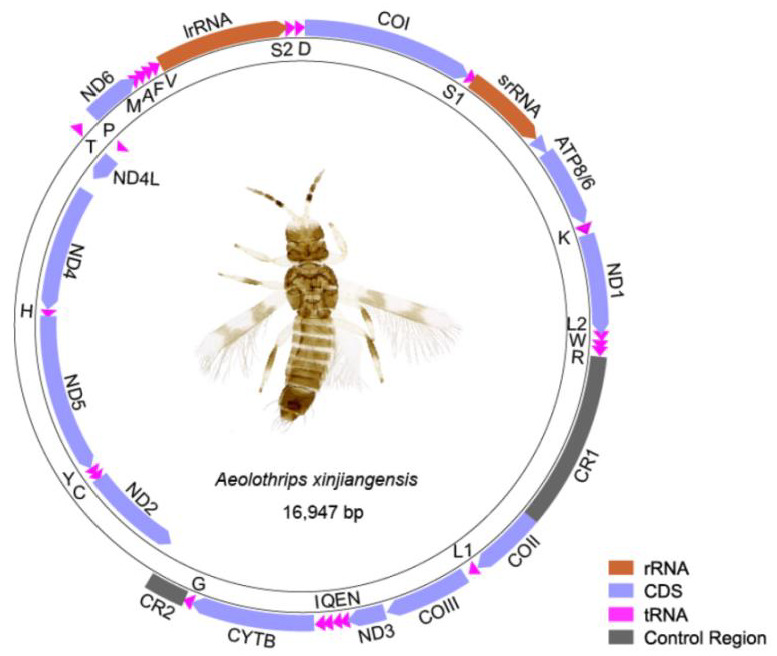
The mitogenome of *Aeolothrips xinjiangensis*. DNA strands are shown as two thick-lined circles. The direction of gene transcription is indicated by the arrows. Abbreviations: *ATP6* and *ATP8*, adenosine triphosphate (ATP) synthase subunits 6 and 8; *COI**–COIII*, cytochrome oxidase subunits 1–3; *CYTB*, cytochrome b; *ND1**–6* and *ND4L*, nicotinamide adenine dinucleotide hydrogen (NADH) dehydrogenase subunits 1–6 and 4L; *srRNA* and *lrRNA*, large and small rRNA subunits. Transfer RNAs are represented by their amino acid abbreviation of the corresponding amino acid, for transfer RNA (L1: CUN; L2: UUR; S1: UCN; S2: AGN).

**Figure 2 insects-13-00585-f002:**
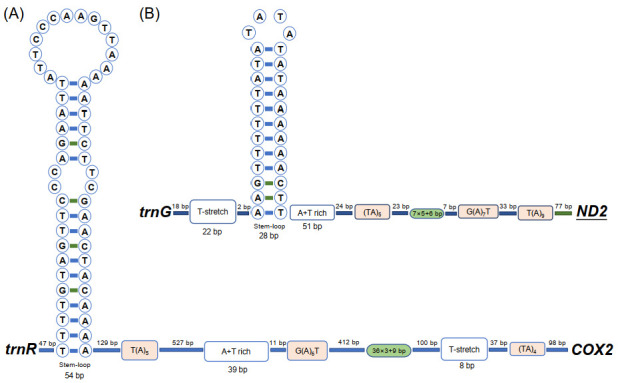
The secondary structure of the two putative control regions of *Aeolothrips xinjiangensis*. (**A**) Control region 1. (**B**) Control region 2. The bases are displayed from left to right following the gene orientation 5′ to 3′. Boxes indicate the motifs; oval frames represent tandem repeats.

**Figure 3 insects-13-00585-f003:**
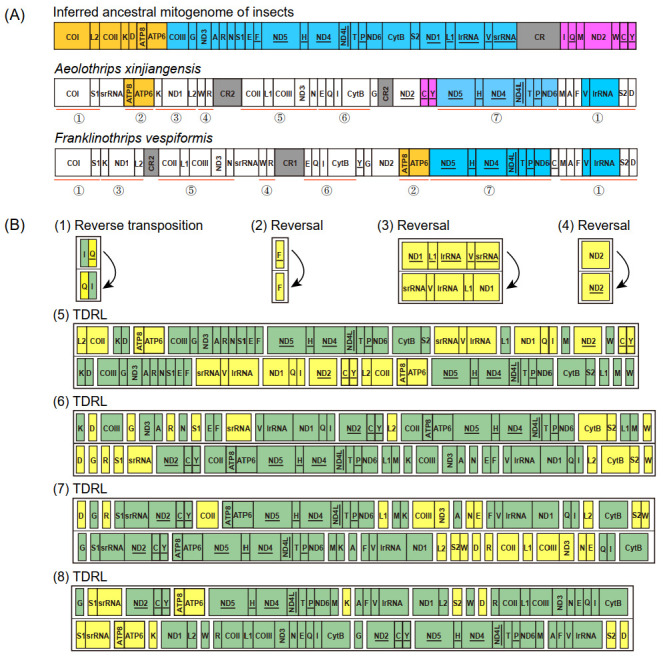
Gene rearrangement. (**A**) Comparison of mitochondrial gene arrangement. The color spray highlights the conserved gene blocks of three mitogenomes. The red line marks seven identical gene blocks between two banded thrips. (**B**) Gene rearrangement events of the mitogenomes from the inferred ancestral mitogenome of insects to *Aeolothrips xinjiangensis* predicted by CREx. In each step, the yellow box highlights the rearranged genes, while the green highlights the conserved genes.

**Figure 4 insects-13-00585-f004:**
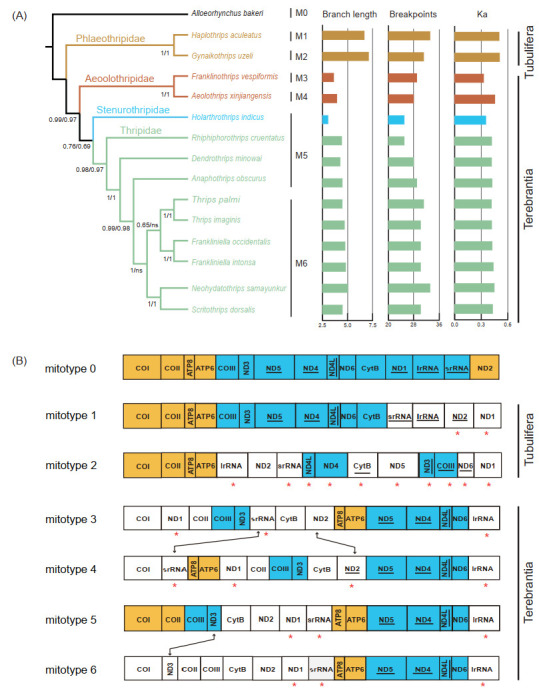
(**A**) The gene rearrangement rates and nucleotide substitution rates of the selected Thysanoptera. Phylogenetic trees were inferred from PhyloBayes. The Bayesian posterior probabilities of the PhyloByayes tree inferred from the PCG and PCG12 datasets were separated by “/”, and the branch lengths are shown on the right side. Different colors indicate data for different families. The mitotypes 0–6 were represented by M0–M6. Ka, the values of the non-synonymous substitution rate. (**B**) Arrangement of the major genes in the mitogenomes used in this study. Mitotype 0 was accessed from the mitogenome of the putative insect ancestor, and the remainder were accessed from the mitogenomes of 14 thrips. Compared to mitotype 0, identical gene blocks are colored, and the inversed gene is highlighted by an asterisk “*”. Gene rearrangement between mitotypes is indicated by double arrow lines.

**Table 1 insects-13-00585-t001:** Taxa used in this study.

Suborder/Order	Family	Species Name	GenBank Number
Hemiptera	Hemiptera	*Alloeorhynchus bakeri*	NC_016432
Terebrantia	Aeolothripidae	*Aeolothrips xinjiangensis* *	MW376485
		*Franklinothrips vespiformis*	MN072395
	Stenurothripidae	*Holarthrothrips indicus*	MN072397
	Thripidae	*Rhiphiphorothrips cruentatus*	MN072396
		*Neohydatothrips samayunkur*	MF991901
		*Anaphothrips obscurus*	NC_035510
		*Thrips palmi*	NC_039437
		*Thrips imaginis*	AF335993
		*Frankliniella intonsa*	JQ917403
		*Frankliniella occidentalis*	NC_018370
		*Scritothrips dorsalis*	NC_025241
		*Dendrothrips minowai*	NC_037839
Tubulifera	Phlaeothripidae	*Haplothrips aculeatus*	NC_027488
		*Gynaikothrips uzeli*	MK940484

* The mitogenome sequenced in the present study.

**Table 2 insects-13-00585-t002:** The correlation between the rate of gene rearrangement in the mitogenomes of thrips and the evolutionary rates of the protein-coding genes in the mitogenomes of thrips.

		Pearson	Spearman
Breakpoints		Ka	Branch length
	*r*	0.563	0.831
	*p* (2-tailed)	0.036	0.000
	*N*	14	14

NOTE. Ka, rate of non-synonymous substitutions; Breakpoints, the rate of gene rearrangement; *r*, correlation coefficient; *p* < 0.05 (significant) and *p* < 0.01 (highly significant); *N*, sample size.

## Data Availability

The annotated mitogenome sequence *Aeolothrips xinjiangensis* has been deposited in the GenBank under the accession number MW376485.

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
