# Peer review of "Positive Correlation of the Gene Rearrangements and Evolutionary Rates in the Mitochondrial Genomes of Thrips (Insecta: Thysanoptera)"

_insects, 2022, doi:10.3390/insects13070585_

Round 1
Reviewer 1 Report
Dear Authors,
I find your paper very interesting and important contribution in the field of Thysanoptera research as well as in the studies of conservation and diversity. I think that your paper will be suitable for publishing after you make revisions below. Also, English editing of the text is mandatory since there are a lot of errors in it.
I have some specific comments and suggestions which I address below:
Line 13: The first sentence of Simple summary is not relevant for the understanding, and it does not incorporate well at this position. Remove this sentence or move it to more appropriate place.
Line 15: Before the latin name Aeolothrips xinjiangensis add “banded thrip species”
Lines 277 – 280: Table 2 needs to be moved up to line 266.
At the end, the Conclusions are missing – I would recommend you add them at the end of the text (after line 280).
Author Response
Responses to Reviewer 1
I find your paper very interesting and important contribution in the field of Thysanoptera research as well as in the studies of conservation and diversity. I think that your paper will be suitable for publishing after you make revisions below. Also, English editing of the text is mandatory since there are a lot of errors in it.
Response: We appreciate the reviewer’s positive evaluation of our work. We have improved our English throughout the manuscript and corrected grammar and spelling mistakes.
I have some specific comments and suggestions which I address below:
Line 13: The first sentence of Simple summary is not relevant for the understanding, and it does not incorporate well at this position. Remove this sentence or move it to more appropriate place.
Response: We removed this sentence according to the reviewer’s comment (Line 78 – Line 79).
Line 15: Before the latin name Aeolothrips xinjiangensis add “banded thrip species”
Response: We added the common name of this species according to the reviewer’s comment (Line 15).
Lines 277 – 280: Table 2 needs to be moved up to line 266.
Response: Changed accordingly (Line 273 – Line 276).
At the end, the Conclusions are missing – I would recommend you add them at the end of the text (after line 280).
Response: We have added a conclusion paragraph at the end of the manuscript (Line 298 – Line 305).
Reviewer 2 Report
In general, the manuscript depicts a very interesting but difficult to study subject, the evolutionary 2 rates. Moreover, the thrip species are among the most destructive crop pests worldwide, fact which gives more credits to the importance of this manuscript.
The molecular and statistical analysis is comprehensible and the tables and figure appear even better the results.
Nevertheless, you can find below some remarks that need the authors‘ attention.
Line 90]
Provide some information about the PCR reagents and conditions, and the appropriate references.
Line 127]
Table 1 contains three genbank numbers that are appearing in the ncbi database with the following message “Record removed. This record was removed by RefSeq staff. Please contact info@ncbi.nlm.nih.gov for further details.”
1) Neohydatothrips samayunkur (NC_039942.1)
2) Thrips imaginis (NC_004371.1)
3) Frankliniella intonsa (NC_021378.1)
Please contact the the RefSeq staff and provide an explanation for this issue or replace the sequences. In the case of the replacement the data should be revalidated or analyzed from the beginning genetically, phylogenetically, and statistically.
Author Response
Responses to Reviewer 2
In general, the manuscript depicts a very interesting but difficult to study subject, the evolutionary rates. Moreover, the thrip species are among the most destructive crop pests worldwide, fact which gives more credits to the importance of this manuscript.
The molecular and statistical analysis is comprehensible and the tables and figure appear even better the results.
Response: We appreciate the reviewer’s positive evaluation of our work.
Nevertheless, you can find below some remarks that need the authors‘ attention.
[Line 90] Provide some information about the PCR reagents and conditions, and the appropriate references.
Response: We have added this information accordingly (Line 93 – Line 101).
[Line 127] Table 1 contains three genbank numbers that are appearing in the ncbi database with the following message “Record removed. This record was removed by RefSeq staff. Please contact info@ncbi.nlm.nih.gov for further details.”
1)Neohydatothrips samayunkur (NC_039942.1)
2)Thrips imaginis (NC_004371.1)
3)Frankliniella intonsa (NC_021378.1)
Please contact the the RefSeq staff and provide an explanation for this issue or replace the sequences. In the case of the replacement the data should be revalidated or analyzed from the beginning genetically, phylogenetically, and statistically.
Response: We have checked the NCBI database and found that the accession numbers for the three species have changed. The mitogenomic sequences of the three species were identical to those used in our previous analyses. We have updated the GenBank numbers for these three thrips in our manuscript Table 1 (Line 143).